# Exploring the Past, Present, and Future of Anti-Angiogenic Therapy in Glioblastoma

**DOI:** 10.3390/cancers15030830

**Published:** 2023-01-29

**Authors:** Ashley B. Zhang, Khashayar Mozaffari, Brian Aguirre, Victor Li, Rohan Kubba, Nilay C. Desai, Darren Wei, Isaac Yang, Madhuri Wadehra

**Affiliations:** 1Department of Neurosurgery, University of California, Los Angeles, CA 90095, USA; 2Sidney Kimmel Medical College, Thomas Jefferson University, Philadelphia, PA 19107, USA; 3Department of Pathology and Laboratory Medicine, University of California, Los Angeles, CA 90095, USA

**Keywords:** angiogenesis, glioblastoma, tumor microenvironment

## Abstract

**Simple Summary:**

Glioblastoma is the most common malignant primary brain tumor. Despite technological advancements and modern therapeutic agents used for treatment, the prognosis remains extremely poor. One unique characteristic of glioblastoma is its highly vascularized nature, enabling the tumors to grow and invade the surrounding brain tissue. This process is known as “angiogenesis” which is critical for growth of glioblastoma cells and has been a topic of interest for scientists. A critical protein that facilitates new blood vessel formation is vascular endothelial growth factor-A (VEGF-A), therefore, bevacizumab, a medication that specifically targets VEGF-A has been approved for treatment of recurrent glioblastoma. However, despite its theoretical potential, bevacizumab has failed to offer significant survival improvement. Furthermore, other agents with mechanisms of action comparable to that of bevacizumab have also not been able to demonstrate favorable results. Herein, we review the current state of anti-vascularization therapeutic agents, and the future of this therapeutic approach.

**Abstract:**

Glioblastoma, a WHO grade IV astrocytoma, constitutes approximately half of malignant tumors of the central nervous system. Despite technological advancements and aggressive multimodal treatment, prognosis remains dismal. The highly vascularized nature of glioblastoma enables the tumor cells to grow and invade the surrounding tissue, and vascular endothelial growth factor-A (VEGF-A) is a critical mediator of this process. Therefore, over the past decade, angiogenesis, and more specifically, the VEGF signaling pathway, has emerged as a therapeutic target for glioblastoma therapy. This led to the FDA approval of bevacizumab, a monoclonal antibody designed against VEGF-A, for treatment of recurrent glioblastoma. Despite the promising preclinical data and its theoretical effectiveness, bevacizumab has failed to improve patients’ overall survival. Furthermore, several other anti-angiogenic agents that target the VEGF signaling pathway have also not demonstrated survival improvement. This suggests the presence of other compensatory angiogenic signaling pathways that surpass the anti-angiogenic effects of these agents and facilitate vascularization despite ongoing VEGF signaling inhibition. Herein, we review the current state of anti-angiogenic agents, discuss potential mechanisms of anti-angiogenic resistance, and suggest potential avenues to increase the efficacy of this therapeutic approach.

## 1. Introduction

Glioblastoma, a WHO grade IV astrocytoma, constitutes approximately half of malignant tumors of the central nervous system (CNS) [1]. Despite technological advancements and aggressive multimodal treatment, prognosis remains dismal. Currently, the median overall survival is 15 months [2], with a five-year survival rate of 6.8% [3]. Although the clinical presentation can vary depending on tumor size and location, patients typically experience headache, nausea/vomiting, seizures, and focal neurological symptoms [2,4]. The characteristics of glioblastoma on magnetic resonance imaging (MRI) have been well-described. A poorly circumscribed tumor, glioblastoma comprises heterogenous intraparenchymal lesions and displays contrast enhancement at the margin, indicative of disruption of the blood–brain barrier [5]. Diagnosis is made on surgical resection or biopsy [5]. Traditionally, the standard of care for glioblastoma includes surgical resection, following by adjuvant radiation therapy and chemotherapy with temozolomide [4].

Over the past decade, angiogenesis, a feature hallmark of this highly vascularized tumor, has emerged as an important target for glioblastoma therapy [6,7]. This review will provide an overview of the current state of anti-angiogenic therapy for glioblastoma and discuss potential avenues for future exploration.

## 2. Angiogenesis

Recognized as one of the most common hallmarks responsible for glioblastoma malignancy, angiogenesis is a mechanism that permits tumor vascularization and infiltration into nearby tissues by endothelial cells, which make up the lining of blood vessels [8]. In response to hypoxia, expression of vascular endothelial growth factor (VEGF) and its receptor (VEGFR) are upregulated, leading to angiogenesis and survival of tumor cells [9]. Moreover, the critical role of microvasculature has been demonstrated in the tumor microenvironment, with findings of cancer stem cells directly participating in vessel formation in glioblastoma by differentiating into endothelial cells or pericytes [10]. As a result, numerous efforts to develop therapy have targeted the VEGF/VEGFR signaling cascade. Most notably, bevacizumab was the first anti-angiogenic agent to be approved for the treatment of several malignancies, including advanced colorectal cancer, advanced non-small-cell lung cancer, and more recently, recurrent glioblastoma [11].

### 2.1. Biology of Angiogenesis in Glioblastoma

In several tumors, including glioblastoma, the high metabolic demand of tumor cells for oxygen and nutrients often exceeds the supply, leading to the presence of hypoxia [12]. Under these conditions, hypoxia-inducible factor (HIF) binds to response elements in the VEGF gene and initiates transcription of VEGF protein [12]. The receptors for VEGF are expressed on the surface of endothelial cells and play a crucial role in angiogenesis by promoting cell proliferation of endothelial cells and tumor cells [12]. In addition to members of the VEGF family, which constitute the most potent pro-angiogenic factors, other key inducers of angiogenesis including fibroblast growth factors and membrane-bound integrins have also been identified [13,14]. 

#### VEGF Signaling Pathway

A soluble protein, VEGF induces a cellular response by binding to its cell-surface receptor, causing receptor dimerization and transphosphorylation [15]. Activation of downstream tyrosine kinase pathway signaling promotes angiogenesis, increased cell motility, and proliferation [15,16,17,18]. During embryonic development, wound healing, and in collateral circulation, VEGF signaling is critical for angiogenesis [19]. In glioblastoma, VEGF maintains this critical proangiogenic role, maintaining the vascular supply that promotes tumor-cell survival [19]. Several anti-angiogenic agents have been designed to target various steps of the VEGF pathway [16,17,20,21,22], thereby inhibiting angiogenesis. These agents have the potential to be incorporated into routine treatment regimen of glioblastoma patients, as studies of pre-clinical models have noted synergistic therapeutic effects when combining anti-VEGF agents with temozolomide, the most commonly used chemotherapeutic agent [23]. 

Furthermore, elevated levels of VEGF activity in glioblastoma can, in turn, lead to abnormal angiogenesis characterized by disorganized and leaky blood vessels [10]. Under physiologic conditions, astrocytes, endothelial cells, and pericytes form a cohesive unit that maintains the integrity of the blood–brain barrier (BBB) [24]. However, the abnormally high VEGF-A expression induces pericyte detachment from the vasculature and results in greatly dilated and enlarged vessels [25]. These pathologic blood vessels provoke disruption of the BBB and are susceptible to microhemorrhages [24]. In glioblastoma, the tumor cells disturb the non-cancerous astrocytes, which in combination with the pathologic vasculature, leads to a disruption in BBB function [26]. This causes an increase in vessel permeability and allows an influx of plasma and fluid into the tumor tissue, causing cerebral edema [26].

The following section will focus on the outcomes of anti-angiogenic agents that were proposed for, or are currently being evaluated in, the treatment of glioblastoma and introduce potential emerging anti-angiogenic therapeutic targets.

## 3. Anti-Angiogenic Therapy in Glioblastoma 

Various anti-angiogenic agents are either currently being used for treatment of glioblastoma or are currently approved for use in other malignancies and are being explored for use in glioblastoma. At the time of writing, there are more than forty ongoing clinical trials in the United States (clinicaltrials.gov, accessed on 23 September 2022) at various stages that are exploring the use of anti-angiogenic agents in glioblastoma. These agents include monoclonal antibodies, receptor fusions proteins, tyrosine kinase inhibitors, and proteasome inhibitors. A summary of the anti-angiogenic agents proposed for treatment of glioblastoma, and a visual illustration of where several of these anti-VEGFR inhibitors act on the VEGF signaling pathway are shown in Table 1 and Figure 1, respectively. 

### 3.1. Anti-VEGF Inhibitors

#### 3.1.1. Bevacizumab

In 2009, the FDA granted accelerated approval for bevacizumab, a humanized anti-VEGF monoclonal antibody, for use in patients with recurrent glioblastoma [27]. This was in response to initial phase II studies exploring the efficacy of bevacizumab in this patient population, which showed a reduction in tumor size, prolongation of progression-free survival, decreased cerebral edema, and improved neurological symptoms [28,29]. Following the results of the phase III, multicenter, randomized study conducted by the European Organization for Research and Treatment of Cancer (EORTC 26101), the FDA granted full approval for treatment of recurrent glioblastoma in 2017.

While this study failed to show a significant increase in overall survival with a bevacizumab-based treatment, progression-free survival was prolonged compared to chemotherapy alone, and bevacizumab promoted a reduced need for corticosteroids [27]. 

As an anti-angiogenic agent, bevacizumab works by specifically binding to circulating VEGF-A ligand and consequently inhibiting its binding to cell-surface receptors [12,30]. In response, there is a decrease in the growth of microvasculature and blood supply to the tumor, and down-regulation of angiogenesis [12,30]. Given the highly vascularized nature of glioblastoma, bevacizumab is theoretically a suitable agent for these tumors, and it remains the most commonly used anti-angiogenic agent in the treatment of recurrent glioblastoma [31,32], despite no significant improve in overall survival [33] due to its role in reducing brain edema [34]. The most common adverse events associated with bevacizumab are gastrointestinal perforations, hemorrhage, and arterial thromboembolism [31]. 

Intriguingly, bevacizumab did not demonstrate an overall survival advantage as part of the first-line treatment for newly diagnosed glioblastoma patients [33,35]. As it stands, the standard of care for this patient population continues to be maximal surgical resection followed by adjuvant radiation and concurrent treatment with temozolomide [35]. 

#### 3.1.2. Aflibercept

Aflibercept is a human recombinant fusion protein with anti-angiogenic properties that serves as a decoy receptor by binding to VEGF-A, VEGF-B, and placental growth factor (P1GF), thus acting as a “VEGF trap” [36]. Structurally, this protein is composed of the second immunoglobulin (Ig) domain of VEGFR-1 and the third domain of VEGFR-2, fused to the constant region (Fc) of human IgG1 [36]. Mechanistically, it has a higher affinity than both VEGFR and bevacizumab for VEGF-A [36]. Furthermore, PlGF has been shown to enhance VEGF signaling activity and mediate angiogenic escape [37], as Batchelor et al. found, levels of P1GF were increased in recurrent glioblastoma patients following treatment with cediranib monotherapy, a VEGFR tyrosine kinase inhibitor [38]. Conceptually, aflibercept should demonstrate increased efficacy in comparison to other anti-angiogenic drugs due to the dual inhibition of both VEGF and P1GF [36,37]. However, phase II clinical trials did not find meaningful improvements in survival of patients with recurrent malignant glioma [37]. The most common adverse events associated with this agent are proteinuria, fatigue, injection-site reactions, and hypertension [36].

#### 3.1.3. Ramucirumab

Whereas the aforementioned agents targeted VEGF, ramucirumab is a human monoclonal antibody that exerts its effect through its high affinity for the extracellular domain of VEGFR-2, blocking its binding to natural ligands [39]. In 2014, ramucirumab received FDA approval as a single-agent treatment for advanced gastric cancer following prior chemotherapy [39]. In a recent non-randomized phase II clinical trial of recurrent glioblastoma patients, ramucirumab was compared to an anti-platelet-derived growth factor receptor (PDGFR) monoclonal antibody, and offered slightly improved progression-free survival and overall survival, with a similar adverse-event profile [40]. Side effects include hypertension, venous thrombosis, diarrhea, and epistaxis [41].

#### 3.1.4. Dovitinib

In addition to inhibiting VEGFR, dovitinib is also a potent inhibitor of basic fibroblast growth factor (bFGF), another pro-angiogenic growth factor that has been shown to be increased in glioblastoma [42,43]. Binding of bFGF to its receptor activates the protein kinase Cα pathway and extracellular signal-regulated kinase (ERK) pathway [43]. In addition, due to the role that bFGF plays in angiogenic escape and bevacizumab resistance in glioblastoma, dovitinib was theorized to offer promise as an anti-angiogenic agent for recurrent glioblastoma [43]. However, the results from a recently published two-arm, phase II clinical trial, compared anti-angiogenic naïve patients with recurrent glioblastoma to patients with glioblastoma progression after prior anti-angiogenic treatment, and dovitinib failed to demonstrate improvements in survival outcomes [43]. Common adverse effects that were reported included appendicitis, fatigue, and thrombocytopenia [43].

### 3.2. Small Molecular Tyrosine Kinase Inhibitors (TKIs)

#### 3.2.1. Sunitinib 

Beyond targeting the VEGF pathway extracellularly, interference of downstream signaling molecules has also been examined [44,45]. Small-molecule TKIs act by reversible, competitive inhibition of adenosine triphosphate (ATP) binding to the tyrosine domain of VEGFRs [41]. Sunitinib is an oral kinase inhibitor of VEGFR, PDGFR, stem-cell-factor receptor (c-KIT), *RET* oncogene tyrosine kinase, FMS-like tyrosine kinase, and colony-stimulating factor-1 receptor [46]. Approved by the FDA as the first anti-VEGF therapy to treat a subset of pancreatic neuroendocrine tumors, this multi-targeted anti-angiogenic TKI blocks downstream signal transduction, thus affecting tumor angiogenesis and growth [47]. The most common adverse reactions include fatigue, diarrhea, nausea, anorexia, vomiting, abdominal pain, hypertension, and thrombocytopenia [47]. Due to its multi-targeted inhibition of angiogenic pathways, sunitinib held promise in glioblastoma therapy. Previously, a phase II study found that single-agent sunitinib therapy in continuous daily dose did not prolong progression-free survival in recurrent glioblastoma [46]. Nevertheless, the STELLAR study, an ongoing multi-center randomized clinical trial, is currently evaluating the efficacy of high-dose, intermittent sunitinib in the treatment of recurrent glioblastoma, compared to lomustine, an alkylating agent of the nitrosourea family capable of permeating the blood–brain barrier [48] (NCT03025893; Table 2). 

#### 3.2.2. Sorafenib 

Similar to sunitinib, sorafenib is another small-molecule TKI with multiple targets including VEGF, PDGFR, and the RAS/RAF/MEK signaling pathways. In a recently published meta-analysis comparing the efficacy and safety between sorafenib and sunitinib as first-line therapy for metastatic renal-cell carcinoma, Deng et al. found that sorafenib did not prolong overall survival as effectively as sunitinib; however, it conferred a lower toxicity [49]. In earlier orthotopic glioblastoma models [50], sorafenib was found to reduce angiogenesis. Following this, a phase II study was conducted to evaluate the efficacy of dual anti-angiogenic therapy with bevacizumab and sorafenib in the treatment of recurrent glioblastoma [51]. While this particular combination did not improve patient outcomes compared to bevacizumab treatment alone, the potential synergistic effect of dual anti-angiogenic therapy simultaneously targeting multiple angiogenic pathways continues to warrant further investigation, as this approach may yield higher clinical efficacy [51]. Common adverse effects of sorafenib therapy include diarrhea, nausea and vomiting, fatigue, rash, and hypertension [49].

#### 3.2.3. Cediranib 

Another multi-kinase inhibitor capable of simultaneously targeting several angiogenic growth factor pathways is cedirnaib, an orally available VEGFR TKI that also targets c-KIT and to a lesser degree, PDGFR [52]. In pre-clinical trials, this small-molecule receptor TKI has shown promising results by reducing microvessel density and metastasis [53]. Additionally, it can be taken orally and is compatible with once-daily dosing due to is half-life of 22 h [45]. However, in a phase III randomized controlled trial, cediranib did not yield any significant improvement in progression-free survival whether in the form of monotherapy or in combination with the chemotherapy agent lomustine, in recurrent glioblastoma patients [45]. Lomustine has been increasingly used as a control arm in clinical trials, in part due to its reputation as the main standard of care for recurrent glioblastoma in Europe where bevacizumab has not been approved, and as such, it remains one of the most widely used drugs, second only to temozolomide, in the treatment of gliomas. [48]. The most commonly reported adverse events include hypertension, dysphonia, fatigue, and diarrhea [53].

#### 3.2.4. Imatinib

A highly selective inhibitor of the tyrosine kinase family, PDGFR, and c-KIT, imatinib has previously been shown to exert anti-angiogenic effects through inhibition of PDGFR [54]. An earlier phase II study evaluating the efficacy and safety of imatinib in combination with hydroxyurea in patients with recurrent meningioma was one of the first to examine combination therapy [55]. This study found that this combination was well-tolerated among patients, and survival outcomes were significantly improved in recurrent meningioma patients with lower-grade tumors. However, recent results published from an open-label, non-randomized phase II trial evaluating imatinib with and without radiotherapy in newly diagnosed or recurrent glioblastoma failed to show an effect [56]. The differences between these two trials may stem from the inherent differences between the tumors which may impact response to treatment. Notably, unlike slow-growing lower-grade meningiomas, glioblastomas represent the most aggressive adult primary brain tumor [55,56]. Common adverse effects that were reported included constipation, fatigue, nausea, and thrombocytopenia [55].

#### 3.2.5. Pazopanib

Pazopanib is another multitargeted TKI of VEGFR, PDGFR, and c-KIT. Due to its ability to target multiple angiogenic pathways, pazopanib was reasoned to exert strong anti-tumor activity [57]. When the efficacy of pazopanib as a single agent in the treatment of recurrent glioblastoma was evaluated in a phase II single-arm study, progression-free survival was not found to be prolonged at a clinically tolerated dose [57]. Previous clinical trials have also evaluated pazopanib in combination with laptinib, a dual TKI of EGFR and HER-2 receptors, in patients with recurrent glioblastoma [58]. While this drug combination was well-tolerated, there was limited evidence of anti-tumor activity [58]. More recently, a phase II trial of oral pazopanib in combination with metronomic topotecan anti-angiogenic therapy for recurrent glioblastoma patients (NCT01931098; Table 2) was completed, which may be able to provide further insight into the value of simultaneously inhibiting several angiogenic pathways in glioblastoma. 

### 3.3. Other Anti-Angiogenic Therapies 

#### 3.3.1. Cilengitide

In preclinical models, integrins αvβ3 and αvβ5, were identified to be implicated in angiogenic pathways and in glioblastoma blood vessels and tumor cells [59,60]. This was the rationale behind using cilengitide, a pentapeptide integrin inhibitor. While early results from phase I and II trials evaluating cilengitide in recurrent glioblastoma were promising and suggested improvement in survival compared to historical controls [61,62], findings from the phase III tCENTRIC trial failed to show improvements in outcome when cilengitide was added to temozolomide, in comparison to the control group [62]. Despite the disappointing outcome, integrins remain a potential valuable target for further review in glioblastoma therapy due to their role in invasion and angiogenesis. 

#### 3.3.2. Marizomib

A recently developed small-molecule proteasome inhibitor, marizomib has been shown to perform a multitude of activities, including induction of apoptosis and down-regulation of cell growth and survival signaling pathways including angiogenesis by interfering with VEGF-dependent migration [63,64]. Compared to other proteasome inhibitors, this irreversible inhibitor is unique in that it has been shown to cross the blood–brain barrier, making it a suitable and attractive agent for brain tumors [65]. Most recently, the results from a phase I/II trial in patients evaluating marizomib alone or in combination with bevacizumab in patients with recurrent glioblastoma were published [65]. Both marizomib monotherapy and dual anti-angiogenic treatment in combination with bevacizumab failed to demonstrate a meaningful benefit. Commonly reported adverse effects include hypertension, confusion, headache, and fatigue [66]. As a relatively new therapeutic agent, more research into marizomib is highly warranted, and there are currently two ongoing clinical trials evaluating marizomib in glioblastoma—a phase II study (NCT03463265; Table 2) and MIRAGE, an international phase III study (NCT03345095; Table 2).

### 3.4. Discovery of Novel Anti-Angiogenic Therapy Targets 

Despite the theoretically suitable mechanism of anti-VEGF therapy and promising preclinical results [7,12,67], these therapeutic agents have failed to produce definitively favorable outcomes in glioblastoma patients [33,37,40,45]. The constantly evolving genetic composition of glioblastoma leads to high rates of intratumor heterogeneity, which subsequently facilitates anti-angiogenic therapy resistance [68,69]. Knowing this, it is evident that targeting the VEGF pathway in glioblastoma via monotherapy is insufficient, and therefore there is an urgent need to discover novel targets that can be used in concert to improve patient survival [32].

Anti-angiogenic therapy can be broken down to two approaches: (1) reducing pro-angiogenic gene expression, and (2) increasing anti-angiogenic gene expression [70]. Apart from VEGF, reduction of other pro-angiogenic factors has also been evaluated [71,72]. For instance, IL-8, a pro-inflammatory cytokine, is involved in increased VEGF expression and signaling [10,70]. Yamanaka et al. reported that retroviral-mediated transfer of antisense IL-8 led to reduced tumor growth, suggesting its potential as a therapeutic target [73]. Conversely, brain angiogenesis inhibitor 1 (BAI1) is an anti-angiogenic protein whose reduced expression has been noted in several malignancies such as colorectal cancer, renal-cell carcinoma, and glioblastoma [70]. Increased expression of BAI1 via recombinant adenovirus in xenograft models led to reduced tumor vasculature [74]. Other notable targets of this category include angiostatin [75,76], endostatin [77,78], and thrombospondin [79,80]. Although the preliminary results have been encouraging, the clinical efficacy of these targets is yet to be determined. 

A recent anti-angiogenic target of interest is epithelial membrane protein-2 (EMP2), a cell-surface protein encoded by growth-arrest-specific 3 (GAS3)/peripheral myelin protein 22 kDa (PMP22) gene family that localizes within the lipid raft domains [80,81,82,83]. Under physiologic conditions, EMP2 appears to stabilize select integrins and modulates their adhesion onto various extracellular matrices [84]. Its expression has been investigated in a number of neoplasms including endometrial carcinoma, breast cancer, and primary brain tumors [32,85,86]. 

As EMP2 is involved in a variety of pathologies including non-cancerous diseases [87,88,89], its signaling mechanism needs further elucidation. Using endometrial cancer xenografts, Gordon et al. demonstrated that modulation of EMP2 expression profoundly changed tumor microvasculature [90]. Under hypoxic conditions, up-regulation of EMP2 promoted VEGF expression through an HIF1α-dependent pathway whereas reduction of EMP2 directly correlated with reduced HIF1α and VEGF expression, supporting its involvement in angiogenesis [90]. More recently, Patel et al. investigated the potential impact of bevacizumab treatment on EMP2 levels in a cohort of 12 glioblastoma patients. In paired analysis, EMP2 histological scores were significantly higher following bevacizumab treatment, and this increase was proportional to the length of treatment [32]. More importantly, patients with higher levels of EMP2 had significantly shorter time to repeat surgery, progression-free survival, and overall survival [32]. Such findings underscore the potential to investigate the clinical implications of EMP2 in glioblastoma. Concurrent evaluation of these proteins along with hypoxia-inducible factors such as HIF1α, or potentially HIF2a, may provide insightful information to better understanding EMP2′s involvement in angiogenesis and designing future therapeutic agents. 

**Table 1 cancers-15-00830-t001:** List of current anti-angiogenic drugs for the treatment of various cancers and their respective study findings in the treatment of glioblastoma.

	Anti-Angiogenic Agent	Main Target(s)	FDA-Approved Indications	Findings in Treatment of Glioblastoma	References
**Anti-VEGF Inhibitors**	Bevacizumab	VEGF-A	Advanced metastatic cancers (lung, colorectal, breast, renal, and recurrent glioblastoma)	Reduction in tumor size, prolongation of PFS, decreased cerebral edema, and improved neurological symptoms in rGBM	[24,25,26]
	Aflibercept	VEGF-A, P1GF	Metastatic colorectal cancer	Phase II clinical trials did not find meaningful improvements in survival of patients with recurrent malignant glioma	[33,34]
	Ramucirumab	VEGFR2	Metastatic NSCLC, gastric cancer, gastroesophageal junction adenocarcinoma, hepatocellular carcinoma	Phase II clinical trial in rGBM showed slightly improved PFS and OS	[36,37]
	Dovitinib	VEGFR, bFGF	-	Phase II clinical trials failed to demonstrate improves in survival in rGBM	[39,40]
**Small Molecular TKIs**	Sunitinib	Multiple RTKs (VEGFR, PDGFR)	GIST, advanced RCC, pNET	Phase II clinical trial found that single-agent sunitinib therapy in continuous daily dose did not prolong PFS *Ongoing Phase II/III clinical trial (NCT03025893)*	[43,44]
	Sorafenib	Multiple RTKs (VEGFR, PDFGR)	Hepatocellular carcinoma, advanced RCC, thyroid carcinoma	Phase II clinical trial in combination with bevacizumab in rGBM did not improve patient outcomes	[46,48]
	Cediranib	VEGFR2	-	Phase III clinical trial found no significant improvement in PFS whether in form of monotherapy or in combination with the lomustine in rGBM	[42,49,50]
	Imatinib	PDGFR	Ph+ CML, Ph+ ALL, MDS/MPD, ASM, recurrent or metastatic DFSP, GIST	Phase II trial evaluating imatinib with and without RT in newly diagnosed or recurrent glioblastoma failed to show an effect	[51,52,53]
	Pazopanib	Multiple RTKs (VEGFR, PDGFR, c-Kit)	Advanced RCC, advanced soft tissue sarcoma	Limited evidence of anti-tumor activity in combination with laptnib *Recently completed Phase II clinical trial (NCT01931098)—awaiting results*	[54,55]
**Integrin Inhibitor**	Cilengitide	Integrins αvβ3 and αvβ5	-	Phase III clinical trial found to show improvements in outcome when cilengitide was added to temozolomide, in comparison to the control group	[56,57,58,59,60]
**Proteasome Inhibitor**	Marizomib	20S proteasome	-	Phase I/II clinical trial in rGBM failed to demonstrate meaningful benefit *Ongoing Phase II and III clinical trials (NCT03463265, NCT03345095)*	[61,62,63,91]

VEGF/R, vascular endothelial growth factor/receptor; P1GF, placental growth factor; bFGF, basic fibroblast growth factor; RTK, receptor tyrosine kinase; c-Kit, stem-cell-factor receptor; NSCLC, non-small-cell lung cancer; GIST, gastrointestinal stromal tumor; RCC, renal-cell carcinoma; pNET, progressive, well-differentiated pancreatic neuroendocrine tumors; Ph+ CML, Philadelphia-chromosome-positive chronic myeloid leukemia; Ph+ ALL, Philadelphia-chromosome-positive chronic myeloid leukemia; MDS/MPD, myelodysplastic/ myeloproliferative diseases; DFSP, dermatofibrosarcoma protuberans; PFS, progression-free survival; rGBM, recurrent glioblastoma; OS, overall survival; RT, radiotherapy. *Italics denote ongoing or recently completed clinical trials*.

**Table 2 cancers-15-00830-t002:** Ongoing clinical trials evaluating anti-angiogenic therapies in glioblastoma.

Clinical Trial Identifier	Treatment	Comparison	Study Type	Study Phase	Study Status *	Primary Endpoints	Estimated Enrolment Number of Patients	Study Start Date	Study Completion Date
**NCT01931098**	Pazopanib + Topotecan	-	Non-randomized, parallel assignment (open label)	II	Completed	6-month PFS, OS, safety	35	29 August 2013	12 September 2019
**NCT03345095 (MIRAGE)**	Marizomib + Temozolomide-RT	Temozolomide-RT	Randomized, parallel assignment (open label)	III	Active, not recruiting	OS, PFS	749	17 November 2017	-
**NCT03463265**	Marizomib + ABI-009	ABI-009	Non-randomized, sequential assignment (open label)	II	Active, not recruiting	Overall response rate, 12-month PFS, OS	56	1 August 2018	-
**NCT03025893 (STELLAR)**	Sunitinib	Lomustine	Randomized, parallel assignment (open label)	II/III	Recruiting	6-month PFS	100	31 August 2018	-

* Status based on https://clinicaltrials.gov/, accessed on 23 September 2022. RT, radiotherapy; PFS, progression-free survival; OS, overall survival; rGBM, recurrent glioblastoma.

## 4. Anti-VEGF-Therapy Resistance 

The highly vascularized nature of glioblastoma led to consideration of anti-angiogenic therapeutic agents for this tumor. A number of such agents have been investigated, and despite promising preclinical data, several clinical trials have thus far failed to demonstrate meaningful improvements in survival [33,37,40,45,92]. Furthermore, a recent meta-analysis noted prolonged progression-free survival but not overall survival with anti-angiogenic therapy in newly diagnosed or recurrent glioblastoma, with or without concurrent chemotherapy [93]. The initial improvement in progression-free survival, particularly with bevacizumab [94] suggests initial susceptibility of glioblastoma tumor cells to anti-VEGF treatment. However, it is highly plausible that over time, other compensatory pathways develop to surpass the effects of anti-VEGF therapy, thus leading to no difference in overall survival [33,94]. 

Other important aspects to consider are the radiographic features, as presence of radiation necrosis has been associated with increased VEGF expression [95]. Furthermore, the “pseudoresponse” phenomenon has been well-described with anti-VEGF agents, as they cause “normalization” of the BBB, reducing the surrounding edema by the tumor. However, this needs to be interpreted with caution as such radiographic changes are likely due to changes in vascular permeability, while true tumor improvement is only marginal [96]. To better design future therapeutic agents, it is crucial to gain a comprehensive understanding of the vascular changes and the molecular mechanisms that drive anti-angiogenic resistance in glioblastoma. One such method involves the analysis of tumor genomics before and after anti-VEGF treatment. Given its frequent use and greatest efficacy amongst anti-VEGF agents, much of the data come from studies that evaluated bevacizumab [94,97]. The intention behind bevacizumab treatment is to starve the cancer cells of VEGF, and subsequently create a hypoxic and unfavorable environment for the tumor [94]. In glioblastoma in vitro models, bevacizumab led to a subsequent up-regulation of proangiogenic factors such as angiogenin and bFGF, both transcriptionally and at the protein level [98]. The same study later analyzed microvasculature density in xenograft mice models which suggested its restoration despite long-term bevacizumab treatment [98]. Such results suggested that glioblastoma tumors do in fact reactivate angiogenesis, likely via up-regulation of other proangiogenic factors such as bFGF, even in the setting of VEGF inhibition. 

As our knowledge of oncogenesis advances, we recognize neoplasms as systemic diseases rather than localized pathologies of organs. Interestingly, the hypoxic environment created as a consequence of anti-angiogenic therapy not only leads to pro-angiogenic factors within the tumor, but also the recruitment of bone-marrow-derived cells (BMDCs) that elicit neo-angiogenesis [97,99]. For example, pro-angiogenic monocytes, including tumor-associated macrophages produce a myriad of proangiogenic cytokines and growth factors that facilitate neo-angiogenesis [100,101]. HIF1α, a transcription factor in the VEGF signaling pathway appears to be an integral component of this recruitment as HIF1α-deficient glioblastoma tumors displayed low BMDCs and severely impaired angiogenesis [101]. The aforementioned studies [94,98,100,101] highlight the complex angiogenic pathway and the intricate molecular signals that drive anti-angiogenic treatment, both locally and systematically. 

Undoubtedly, elucidating the mechanisms of anti-angiogenic resistance in glioblastoma remains a critical challenge and a much-needed endeavor [97]. However, our current understanding has proven that anti-angiogenic evasion is different from classic drug resistance seen in traditional therapy. In contrast to chemotherapy resistance, in which there is acquisition or selection of gene mutation in the drug target, bevacizumab evasion consists of adaptive changes that upregulate other angiogenic markers [98,100,101], despite continued inhibition of VEGF [97].

## 5. Future Directions and the Role of Combination Therapy

Given the convincing evidence that anti-VEGF monotherapy does not effectively improve survival in glioblastoma patients [22,93], it is imperative to assess the efficacy of multimodal anti-angiogenic therapies. Peterson et al. demonstrated improved survival of murine glioblastoma models when utilizing dual anti-angiogenic therapy by targeting VEGFR and angiopoietin-2 (Ang-2) [17]. Such an approach is also a work in progress in other malignancies [102,103]. There is compelling evidence that EMP2 is a suitable target to antagonize [32,104] in combination with bevacizumab, or other anti-angiogenic agents. However, other proteins in the VEGF pathway including VEGFR tyrosine kinases may be reasonable options as well. In the same vein as this line of exploratory research for glioblastoma, combination therapy with anti-angiogenic therapy and immunotherapy, which has proven successful in renal-cell carcinoma [13] and non-small-cell lung cancer, may be a viable option [105]. Regardless, the future of anti-angiogenic therapy needs to focus on interfering with this signaling pathway from two or more angles, as monotherapy will likely not result in substantial improvements in survival outcomes.

## 6. Conclusions

Glioblastoma is the most common malignant primary brain tumor. Despite advances in therapeutics, prognosis remains extremely poor. A characteristic hallmark of this highly vascular tumor, angiogenesis, has increasingly become an important target for therapy. Nevertheless, current literature suggests that anti-angiogenic monotherapy with bevacizumab and other agents does not produce favorable results. Therefore, it is imperative to delineate the molecular mechanisms of anti-angiogenic resistance and the interplay between these agents, the tumor cell microenvironment, and angiogenic signaling pathways. Going forward, the goal remains to identify novel targets that can be effectively utilized in dual or multimodal anti-angiogenic therapy to ultimately improve clinical outcomes and patient survival for this highly aggressive tumor. 

## Figures and Tables

**Figure 1 cancers-15-00830-f001:**
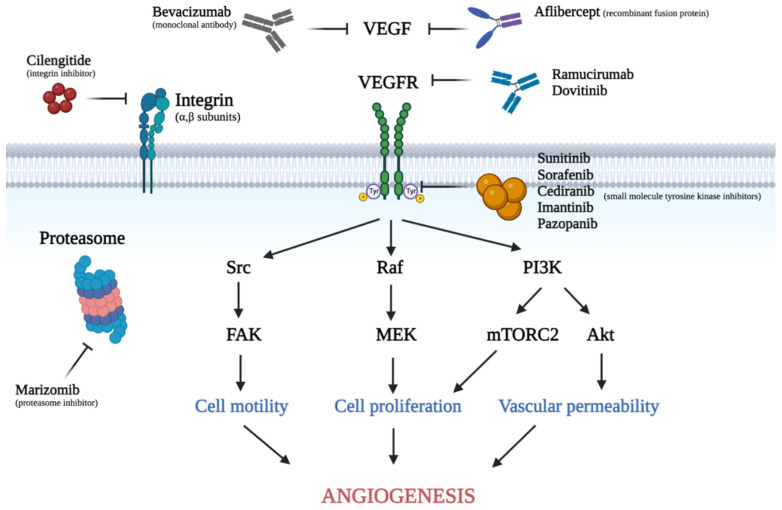
Visual demonstration of the VEGF signaling pathway and various anti-angiogenic agents discussed in this review. Soluble VEGF binds to its receptors located on the cell surface. This interaction leads to receptor dimerization and phosphorylation of tyrosine residues, activating the downstream signaling cascade such as Src, Raf, and PI3K. This eventually causes increased cell motility, cell proliferation, and vascular permeability. This figure was created with Biorender.com. VEGF: vascular endothelial growth factor, VEGFR: vascular endothelial growth factor receptor, PlGF: placental growth factor, TKI: tyrosine kinase inhibitor.

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
