# Peer review of "Exploring the Past, Present, and Future of Anti-Angiogenic Therapy in Glioblastoma"

_cancers, 2023, doi:10.3390/cancers15030830_

Round 1
Reviewer 1 Report
Zhang and colleagues wrote a comprehensive review on medications targeting molecules, such as VEGF-A, to inhibit angiogenesis for recurrent glioblastoma multiform (GBM). This angiogenic activity observed in GBM contributes to its infiltration into surrounding tissues. Unfortunately, bevacizumab, and well as other anti-angiogenic agents against VEGF-A and other targets in angiogenesis failed to improve patient prognosis.
The authors are encouraged to include in their review the ability of radiation to increase VEGF secretion by GBM cells in vitro and in animal models. Since radiotherapy is included in the standard treatment of GBM, how it could interfere with anti-angiogenic medications deserves to be better studied in preclinical models. A large volume of healthy tissue around the surgical cavity is included in the irradiated volume because it may have been infiltrated by GBM cells. A local increase in VEGF induced by radiotherapy can therefore promote their proliferation as well as the angiogenesis which could contribute to disease recurrence. In addition, it might be appropriate to question the sequence of treatments. Determining when to administer an anti-angiogenic drug versus radiotherapy could be a game-changer.
Author Response
Thank you for the kind reviews, please find a point-by-point response in italics below.
Thank you for your insightful comment as it raises an interesting point. For the purposes of this review, we aimed to focus our review on the process of angiogenesis as a hallmark of glioblastoma and subsequently, the current state of anti-vascularization agents and the future of this therapeutic approach. Although examining the relationship and interplay between radiotherapy and anti-angiogenic therapies will offer more insightful information, doing so was outside the scope of this review. We will endeavor to incorporate a literature review of radiotherapy in glioblastoma in future reviews.
Reviewer 2 Report
In this review article, the authors summarized the past and current situation of anti-angiogenic therapy on glioblastoma, and suggested several possible trials in the future. The article is mostly well written, and covered well most of the possible candidate molecules targeting angiogenesis in this highly malignant tumor.
The reviewer has a few comments.
1, BBB (blood brain barrier) might be one of the difficult points to obtain a continuous clinical effect in this glioblastoma therapy. Even the anti-VEGF proteins such as Bevacizumab could suppress highly permeabilized and edematous tumor vessels at early phase of therapy, the blocking of permeability and new angiogenesis might reconstruct the BBB surrounding tumor tissues, which may suppress the efficacy of anti-angiogenic proteins due to a lower permeability of antibodies/anti-VEGF proteins towards tumor tissues. This might be some difference from other solid tumors such as colorectal cancer and lung cancer. It seems a little better to add more on BBB in Discussion or other parts.
2, Anti-angiogenic therapy might be effective to early stage of anti-glioblastoma therapy, but including reconstruction of BBB, efficacy of anti-angiogenic drugs might decrease, and due to the double stress of hypoxia and low nutrition, tumor cells may acquire more malignant phenotype. Thus, dual or multimodal anti-angiogenic therapy in the future might be better (as a possibility) to have a stepwise therapy such as, at the first stage (for example, 2-3 months) with dual (anti-angiogenic drug plus anti-tumorigenic drug) therapy, then, at the next phage, triple (anti-angiogenic drug plus additional anti-tumorigenic drugs) therapy to overcome the more malignant phenotype expected at the end of first stage. How do the authors consider such a way?
3, in the Figure: VEGFR2, the major signal transducer tor vascular endothelical proliferation, was shown to utilize a unique pathway: VEGFR2-PLCg-PKC pathway towards Raf-MEK, which is a relatively mild proliferation pathway different from other typical oncogenic tyrosine kinase signaling, using Ras-pathway (Takahashi et al. EMBO J. 20, 2768-2778, 2001, and other papers). It might be related with tumor resistance against anti-angiogenic drugs.
4, Page 4, line 151; PlGl : is it a mistype ?
Author Response
Thank you for the kind reviews, please find a point-by-point response in italics below.
Comments from the editors and reviewers:
Reviewer 2: In this review article, the authors summarized the past and current situation of anti-angiogenic therapy on glioblastoma, and suggested several possible trials in the future. The article is mostly well written, and covered well most of the possible candidate molecules targeting angiogenesis in this highly malignant tumor. The reviewer has a few comments.
1, BBB (blood brain barrier) might be one of the difficult points to obtain a continuous clinical effect in this glioblastoma therapy. Even the anti-VEGF proteins such as Bevacizumab could suppress highly permeabilized and edematous tumor vessels at early phase of therapy, the blocking of permeability and new angiogenesis might reconstruct the BBB surrounding tumor tissues, which may suppress the efficacy of anti-angiogenic proteins due to a lower permeability of antibodies/anti-VEGF proteins towards tumor tissues. This might be some difference from other solid tumors such as colorectal cancer and lung cancer. It seems a little better to add more on BBB in Discussion or other parts.
Thank you for your comment. We have updated the manuscript to reflect the suggested changes and have added more on BBB in the Introduction, which can be found on page 3, lines 97 – 107, “Furthermore, elevated levels of VEGF activity in glioblastoma can in turn, lead to abnormal angiogenesis characterized by disorganized and leaky blood vessels 10. Under physiologic conditions, astrocytes, endothelial cells, and pericytes form a cohesive unit that maintains the integrity of the blood-brain barrier (BBB)24. However, the abnormally high VEGF-A expression induces pericyte detachment from the vasculature and results in greatly dilated and enlarged vessels25. These pathologic blood vessels provoke disruption of the BBB and are susceptible to microhemorrhages 24. In glioblastoma, the tumor cells disturb the non-cancerous astrocytes, which in combination with the pathologic vasculature, leads to a disruption in BBB function26. This causes an increase in vessel permeability and allows an influx of plasma and fluid into the tumor tissue, causing cerebral edema26.”
2, Anti-angiogenic therapy might be effective to early stage of anti-glioblastoma therapy, but including reconstruction of BBB, efficacy of anti-angiogenic drugs might decrease, and due to the double stress of hypoxia and low nutrition, tumor cells may acquire more malignant phenotype. Thus, dual or multimodal anti-angiogenic therapy in the future might be better (as a possibility) to have a stepwise therapy such as, at the first stage (for example, 2-3 months) with dual (anti-angiogenic drug plus anti-tumorigenic drug) therapy, then, at the next phage, triple (anti-angiogenic drug plus additional anti-tumorigenic drugs) therapy to overcome the more malignant phenotype expected at the end of first stage. How do the authors consider such a way?
Thank you for your comment. We aimed to focus on our review on highlighting the current state of anti-angiogenic therapy, which has predominately, to date, focused on single-agent anti-angiogenic therapy. We agree with your insightful comments that dual or multi-modal therapy may overcome more malignant phenotypes and we have aimed to discuss this approach in the section “Future Directions and the Role of Combination Therapy,” on page 14, lines 507-522. In addition, Table 2 highlights the ongoing clinical trials, which include three current trials that invovle dual therapy (anti-angiogenic drug plus radiotherapy or another anti-tumorigenic drug).
3, in the Figure: VEGFR2, the major signal transducer tor vascular endothelical proliferation, was shown to utilize a unique pathway: VEGFR2-PLCg-PKC pathway towards Raf-MEK, which is a relatively mild proliferation pathway different from other typical oncogenic tyrosine kinase signaling, using Ras-pathway (Takahashi et al. EMBO J. 20, 2768-2778, 2001, and other papers). It might be related with tumor resistance against anti-angiogenic drugs.
Thank you for your comment. We hope that our Section on “Anti-VEGF Therapy Resistance,” on page 13, lines 453 – 505 explores this topic of resistance in glioblastoma and discusses the considerations and challenges that resistance poses to anti-angiogenic therapy.
4, Page 4, line 151; PlGl : is it a mistype ?
Thank you for your comment. We have updated the manuscript to reflect this edit, which can be found on page 4, lines 162 - 165, “Furthermore, PlGF has been shown to enhance VEGF signaling activity and mediate angiogenic escape [34], as Batchelor et al., found that levels of P1GF were increased in recurrent glioblastoma patients following treatment with cediranib monotherapy, a VEGFR tyrosine kinase inhibitor [35].”